# Long-term mortality in mothers of infants with neonatal abstinence syndrome: A population-based parallel-cohort study in England and Ontario, Canada

Astrid Guttmann[1,2,3,4,5]☯*, Ruth Blackburn[6]☯, Abby Amartey[1], Limei Zhou[1], Linda Wijlaars[7], Natasha Saunders[1,2,3,4,5], Katie Harron[7], Maria Chiu[1,4], Ruth Gilbert[7]

**1** ICES, Toronto, Ontario, Canada, **2** Hospital for Sick Children, Toronto, Ontario, Canada, **3** Department of Paediatrics, University of Toronto, Toronto, Ontario, Canada, **4** Institute of Health Policy, Management and Evaluation, University of Toronto, Toronto, Ontario, Canada, **5** Dalla Lana School of Public Health, University of Toronto, Toronto, Ontario, Canada, **6** UCL Institute of Health Informatics, London, United Kingdom, **7** UCL Great Ormond Street Institute of Child Health, London, United Kingdom

☯ These authors contributed equally to this work.

* astrid.guttmann@ices.on.ca

**Data Availability Statement:** Ontario statement: Data used for this study were housed at ICES, an independent not-for-profit corporation. The data

## Abstract

### Background

Opioid addiction is a major public health threat to healthy life expectancy; however, little is known of long-term mortality for mothers with opioid use in pregnancy. Pregnancy and delivery care are opportunities to improve access to addiction and supportive services. Treating neonatal abstinence syndrome (NAS) as a marker of opioid use during pregnancy, this study reports long-term maternal mortality among mothers with a birth affected by NAS in relation to that of mothers without a NAS-affected birth in 2 high-prevalence jurisdictions, England and Ontario, Canada.

### Methods and findings

We conducted a population-based study using linked administrative health data to develop parallel cohorts of mother–infant dyads in England and Ontario between 2002 and 2012. The study population comprised 13,577 and 4,966 mothers of infants with NAS and 4,205,675 and 929,985 control mothers in England and Ontario, respectively. Death records captured all-cause maternal mortality after delivery through March 31, 2016, and cause-specific maternal mortality to December 31, 2014. The primary exposure was a live birth of an infant with NAS, and the main outcome was all deaths among mothers following their date of delivery. We modelled the association between NAS and all-cause maternal mortality using Cox regression, and the cumulative incidence of cause-specific mortality within a competing risks framework. All-cause mortality rates, 10-year cumulative incidence risk of death, and crude and age-adjusted hazard ratios were calculated. Estimated crude 10-year mortality based on Kaplan–Meier curves in mothers of infants with NAS was 5.1% (95% CI 4.7%–5.6%) in England and 4.6% (95% CI 3.8%–5.5%) in Ontario versus 0.4% (95% CI 0.41%–

set from this study is held securely in coded form at ICES. While data sharing agreements prohibit ICES from making the data set publicly available, access may be granted to those who meet pre-specified criteria for confidential access. Information about the application process, including the DAS Data Request Form and the criteria for access, including, for example, confirmation of approval by a Research Ethics Board, are available at https://www.ices.on.ca/DAS/Submitting-your-request. For general information visit www.ices.on.ca/DAS or email das@ices.on.ca. England statement: Authors do not have permission to share patient-level HES data. HES data are available from the NHS Digital Data Access Request Service team (enquiries@nhsdigital.nhs.uk) for researchers who meet the criteria for access to confidential data.

**Funding:** Ontario: This study was supported by the ICES, which is funded by an annual grant from the Ontario Ministry of Health and Long-Term Care (MOHLTC). AG was supported by a Canadian Institute for Health Research Applied Chair in Reproductive and Child Health Services and Policy Research (Funding Reference Number: APR 126377) [http://www.cihr-irsc.gc.ca/e/46349.html]. England: This study/project was funded by the National Institute for Health Research (NIHR) Policy Research Programme and supported by the Administrative Data Research Centre for England by the Economic and Social Research Council (grant reference number ES/L007517/1) [https://esrc.ukri.org/research/our-research/administrative-data-research-uk/]; the Farr Institute of Health Informatics Research London [http://farrinstitute.org/], funded by the Medical Research Council and 7 other funders (grant no MR/K006584/1); RB for a UKRI Innovation Fellowship [https://www.ukri.org/] funded by the Medical Research Council (Grant No: MR/S003797/1). This research benefits from and contributes to, but was not commissioned by, the National Institute for Health Research Children and Families Policy Research Unit. We acknowledge support from the NIHR GOSH Biomedical Research Centre. This work was supported by Health Data Research UK, an initiative funded by UK Research and Innovation, Department of Health and Social Care (England) and the devolved administrations, and leading medical research charities. The funders had no role in study design, data collection and analysis, preparation of the manuscript or decision to publish.

**Competing interests:** The authors have declared that no competing interests exist.

**Abbreviations:** NAS, neonatal abstinence syndrome; NHS, National Health Service.

0.42%) in England and 0.4% (95% CI 0.38%–0.41%) in Ontario for controls (*p* < 0.001 for all comparisons). Survival curves showed no clear inflection point or period of heightened risk. The crude hazard ratio for all-cause mortality was 12.1 (95% 11.1–13.2; *p* < 0.001) in England and 11.4 (9.7–13.4; *p* < 0.001) in Ontario; age adjustment did not reduce the hazard ratios. The cumulative incidence of death was higher among NAS mothers than controls for almost all causes of death. The majority of deaths were by avoidable causes, defined as those that are preventable, amenable to care, or both. Limitations included lack of direct measures of maternal opioid use, other substance misuse, and treatments or supports received.

## Conclusions

In this study, we found that approximately 1 in 20 mothers of infants with NAS died within 10 years of delivery in both England and Canada—a mortality risk 11–12 times higher than for control mothers. Risk of death was not limited to the early postpartum period targeted by most public health programs. Policy responses to the current opioid epidemic require effective strategies for long-term support to improve the health and welfare of opioid-using mothers and their children.

## Author summary

### Why was this study done?

- Opioids are now a leading cause of death of young and middle-aged people in North America, and rates of use and misuse are also increasing in the United Kingdom.

- There has been a steep increase in the number of women who use opioids in pregnancy, which often results in their infants having signs of withdrawal, called neonatal abstinence syndrome (NAS).

- Most studies of NAS focus on the child's health, with very few about the mother's health.

- Pregnancy is an opportunity to identify mothers who may need addiction services and other support to improve their health and that of their families.

- We wanted to measure how much more commonly women whose infants had NAS die from all causes in the years following birth compared with other mothers in both Canada and England.

### What did the researchers do and find?

- We used population-based data that included all hospital births in England and Ontario, Canada, from 2002 to 2012 and analyzed death rates through to 2016.

- We studied 13,577 mothers in England and 4,966 in Ontario with infants with NAS and 4,205,675 mothers in England and 929,985 in Ontario whose infants did not have NAS.

- Mothers with infants with NAS were more likely than mothers of infants without NAS to live in poverty, have other mental health and addiction problems, and have their infants placed in out-of-home care.

- At 10 years after giving birth, 5.1% of English mothers with infants with NAS and 4.6% of Ontarian ones had died, compared with 0.4% of mothers whose infants did not have NAS in both countries. This translates to an 11–12 times higher risk of death associated with prenatal opioid use.

- The majority of deaths in mothers with infants with NAS were from avoidable causes such as intentional and unintentional injuries.

- We did not observe any particular time after birth that was associated with a high risk of death.

### What do these findings mean?

- Women whose infants have NAS are at much higher risk of dying in the years following birth than mothers whose infants did not have NAS.

- Clinicians should ensure that mothers of infants with NAS receive available services to support their health.

- Researchers need to test which models of care can best be used to improve the health of these mothers and reduce the risks they face that may cause them to die early.

- Policymakers focused on harm reduction related to opioid use should include a focus on pregnant mothers and their children; programs will need to extend past the current typical period of only 1–2 years after birth.

## Introduction

Opioid use is responsible for an important increase in premature mortality in young and middle-aged adults in the US [1] and Canada [2], 2 of the countries with the highest per capita prescription opioid consumption in Western industrialized nations [3]. Other countries such as England have seen similar rates of increase in prescription opioid use but not concomitant increases in mortality rates, likely related in part to better access to addiction treatment and more oversight of prescription opioids [4]. Across all of these jurisdictions, there is increasing opioid use by pregnant women, and while little is known about associated maternal mortality, a recent study using data from 22 US states and the District of Columbia reports a higher than 3-fold increase from 2007 to 2017 in opioid-related deaths in women during or within the first year after pregnancy [5].

Population-based surveillance of opioid use during pregnancy is difficult given the lack of prescription medication data in many jurisdictions and the challenges in measuring illicit use. Neonatal abstinence syndrome (NAS) is coded in the infant birth hospitalization record and offers a widely used but imperfect proxy measure of maternal opioid use during pregnancy. NAS manifests typically within hours to 1 to 2 days of delivery with autonomic,

gastrointestinal, and neurologic symptoms of drug withdrawal in the infant that often require prolonged postnatal care [6]. Not all infants exposed to opioids in utero will experience NAS [7,8]. In trial and observational settings, approximately half of women receiving methadone or buprenorphine maintenance therapy in pregnancy gave birth to an infant with signs of NAS [7], and estimates of up to 91% have been reported in other groups of women with chronic opioid use [6]. The incidence of NAS rose dramatically between the early 2000s and 2014 in the US (from 28 to 144 per 10,000 births) [9] and Canada (from 18 to 54 per 10,000 live births) [10], but remained relatively stable in Australia and England [11,12].

Mothers of infants with NAS represent a heterogeneous group including those using prescription opioids and opioid agonists for analgesia, those on medically supervised maintenance treatment for dependence, and those using illicit opioids [12]. Overall social disadvantage and opioid use are inextricably linked, and many women who misuse these drugs are at greater risk of adversity, including deprivation, violence and abuse, and use of other substances [13]. All of these factors impact negatively on maternal health and may diminish parenting capacity [14]. Improving maternal health and preventing premature maternal mortality in this population is therefore critical for both mothers and their children. Pregnancy and delivery care are opportunities to improve access to addiction and supportive services. While public health programs such as nurse home visits tend to focus on pregnancy and the early postpartum period [15], it is largely unknown whether this is the only period of risk for poor outcomes for mothers who use opioids in pregnancy.

Evidence on mortality for mothers who use opioids in pregnancy is limited but consistently shows increased rates around the time of delivery (Table 1) [5,16–18]. Very high rates of longer-term maternal mortality have been reported in 2 older studies (1 Australian and 1 Finnish) [19,20], but these studies may not reflect the current opioid epidemic in North America or the UK [4]. Recent estimates of mortality from a meta-analysis of people with substance misuse disorder and homeless and prison populations reported a standardized mortality ratio for women of 11.9 (95% CI 10.4–13.3), which was higher than the equivalent figure for men (7.9; 95% CI 7.0–8.7) [21]. There is a dearth of information about long-term health outcomes for women—particularly mothers—with opioid use, which is an important knowledge gap given rising rates of prescription and illicit opioid use. Pregnancy can be seen as a window of opportunity for identifying and managing substance misuse and the implications for parenting capacity.

In this study, we capitalize on linked population-based maternal–infant healthcare records and mortality files in 2 jurisdictions (England and Ontario, Canada). Both have similar healthcare systems, including universal access to healthcare and similar postnatal public health programs that focus predominantly on the year after birth. We hypothesized that NAS mothers would have significantly higher rates of mortality than control mothers. We aimed to quantify this excess mortality, investigating all-cause mortality as the primary outcome and cause-specific mortality as secondary outcomes. We present all-cause mortality in relation to some key maternal characteristics at the time of birth, to identify clinically useful sub-groups with poor prognostic outcomes and to guide opioid misuse policy and research.

## Methods

An analytic plan was written and approved before starting statistical analyses (S1 Text). All analyses were undertaken as planned and reported, with the exception of the removal of comorbidity adjustment in the Cox proportional hazards model (see section on statistical analysis below) as a result of feedback from peer review. This study is reported as per the

**Table 1. Mortality rates from 6 population-based studies of women with substance misuse during pregnancy.**

| Study (country) | Study design (study years) | Study population | Duration of follow-up | Number of participants | Number of deaths | Mortality rate | Mortality rate ratio |
|---|---|---|---|---|---|---|---|
| *Long-term mortality* | | | | | | | |
| Kahila et al. 2010 [20] (Finland) | Registry-based retrospective case–control (1992–2001) | Women living in Helsinki metropolitan area who gave birth between 1992 and 2001 and were referred to a specialist alcohol/substance misuse antenatal clinic | Mean of 9.4 (cases) and 10.1 (controls) years | 2,316 (524 cases) | 46 (42 cases) | 8.52 (cases) and 0.22 (controls) per 1,000 person-years | OR: 38 (95% CI 14–108) |
| Hser et al. 2012 [22] (US) | Prospective cohort study (2000–2010) | Pregnant or parenting women assessed and admitted to 40 drug treatment programs in California between 2000 and 2002 | 8–10 years | 4,447 | 194 | 4.47 per 1,000 person-years | SMR: 8.4 (95% CI 7.2–9.6) |
| *Short-term mortality* | | | | | | | |
| Wolfe et al. 2005 [18] (US) | Linked discharge, birth, and death cohort (1991–1998) | Mother and newborn pairs between 1991 and 1998 in California with drug and/or alcohol use indication as discharge diagnostic codes during pregnancy | Death ≤ 72 hours after delivery | 4,536,701 (54,290 cases) | 1,944 (62 cases) | Unknown/not stated | RR: 2.7 (95% CI 2.1–3.5) |
| Whiteman et al. 2014 [17] (US) | Cross-sectional analysis (1998–2009) | Women with a pregnancy-related hospital discharge between 1998 and 2009 with indication of opioid use during pregnancy on the discharge record | Death during the delivery hospital stay | 55,781,965 (138,224 cases) | Unknown/not stated | 0.8 (cases) and 0.1 (controls) per 1,000 pregnancy-related discharges | OR: 5.9 (95% CI 3.7–9.3) |
| Maeda et al. 2014 [16] (US) | Cross-sectional analysis (1998–2011) | Women with a delivery admission between 2007 and 2011 with indication of opioid abuse or dependence during pregnancy on the discharge record | Death during the delivery hospital stay | 20,517,479 (60,994 cases) | 1,331 (20 cases) | 0.03 (cases) and 0.006 (controls) per 100 delivery hospitalizations | Adjusted OR: 4.6 (95% CI 1.8–12.1) |
| Gemmill et al. 2019 [5] (US) | Retrospective cohort study (2007–2016) | Women aged 15–49 years with a pregnancy-associated death between 2007 and 2016 across 22 US states and the District of Columbia | Death while pregnant or within 1 year of end of pregnancy | Unknown/not stated | Unknown/not stated | Mortality per 100,000 live births: 31.7 in 2007 to 42.3 in 2016 (all-cause); 1.3 in 2007 to 4.2 in 2016 (opioid-related) | Unknown/not stated |

Non-population-based studies [19,23–25] or studies that examined opioid use but not during pregnancy [26–29] were excluded.

OR, odds ratio; RR, relative risk; SMR, standardized mortality ratio.

Strengthening the Reporting of Observational Studies in Epidemiology (STROBE) guidelines (S2 Text).

## Cohort

We derived whole-region population-based cohorts of mothers aged 12 to 49 years and their live born infants delivered between April 1, 2002, and December 31, 2012, in England and Ontario, Canada. This period coincides with a steep increase in opioid use, particularly in Canada. The cohorts used longitudinal hospital discharge records for mothers (back to April 1, 1997, for some covariates) linked to hospitalization records for the infant. Mortality registration records were linked to hospital records for the mother and infant. In England, de-identified data on inpatient admissions for all National Health Service (NHS) hospitals linked to Office for National Statistics mortality data were obtained from NHS Digital, linked between mother and infant using previously reported methods [30], and analyzed within the UCL Data

Safe Haven, England [31]. In Ontario, hospital discharge records were obtained from the Canadian Institute for Health Information Discharge Abstract Database and the Ontario Mental Health Reporting System. Cause of death information was taken from the Office of the Registrar General's vital statistics database (data available only until 2014), and demographic information captured from the 2006 Canadian Census and the Registered Persons Database (which includes date of death). Canadian datasets were linked using unique coded identifiers common across all aforementioned datasets and analyzed at ICES in Toronto, Ontario, Canada. We restricted the cohort to singleton births, and if a woman had more than 1 live birth delivery during the study period, 1 delivery was chosen at random as the focus of the study, i.e., a delivery date was selected at random and used as the entry point for the mother (referred to as the index delivery), and all other deliveries were ignored.

In both jurisdictions, infants with a diagnosis of NAS were identified using International Classification of Diseases and Related Health Problems—10$^{th}$ Revision (ICD-10) codes P96.1 (neonatal withdrawal symptoms from maternal use of drugs of addiction) or P04.4 (newborn [suspected to be] affected by maternal use of drugs of addiction) recorded during the delivery admission or subsequent readmission within 14 days of birth [11]. The ICD-9 equivalent to P96.1 has been shown to have high sensitivity (88.1%; 95% CI 83.3%–91.7%) and specificity (97.0%; 95% CI 93.8%–98.5%) and a positive predictive value of 91.2% (95% CI 86.8%–94.2%) for measuring NAS [32] and is used by the US Agency for Healthcare Research and Quality [33]. We included P04.4 as it is often used for opioid withdrawal in both jurisdictions. In a sample of all Ontario women giving birth in hospital from 2014 to 2017 ($n$ = 464,400), during which time all prescription opioids were registered, over 56% of women whose infant had a diagnosis of P04.4 had either a prescription for opioids or opioid agonists, or a healthcare encounter related to opioid use during pregnancy (personal communication, A. Camden, University of Toronto, September 4, 2019). In our study, NAS was ascertained by P96.1 in 83% of cases in England and 65.7% in Ontario. The English maternal cohort was identified using linked data for singleton babies and mothers, corresponding to 96% of all live births in England within the study period. Mothers and babies were matched deterministically using data on hospital, general practitioner practice, maternal age, birthweight, gestation, birth order, and sex, or probabilistically using additional data including admission dates, ethnicity, and partial postal code [34]. Ontario mothers were identified using a unique number linking all newborn and maternal hospital records that is assigned at the delivery hospitalization.

## Outcomes and covariates

The main outcome was all-cause mortality measured from April 1, 2002, to March 31, 2016, derived from linked death registrations. Cause-specific mortality was available only up until December 31, 2014, for Ontario. Cause-specific deaths were classified as avoidable (defined as preventable, amenable to care, or both), unavoidable (as specified by the UK's Office for National Statistics) [35], or cancer (avoidable and unavoidable).

Longitudinal hospital records were used to derive baseline characteristics at the index delivery including maternal age (12–19 years, 20–34 years, and 35+ years); time since last birth (defined as no previous births [with lookback to April 1, 1997, only], <2 years, 2–5 years, and 6+ years); neighbourhood income quintile; urban or rural residence; pregnancy-related outcomes: cesarean delivery, gestational age at delivery (<34 weeks, 34–36 weeks, and 37+ weeks), gestational hypertension, pre-eclampsia/eclampsia, and gestational diabetes; neonatal mortality (death within 28 days of birth); and infant discharge from hospital to social services. We excluded observations with extreme values, including women aged <12 years or >49 years at the date of delivery. Missing values for neighbourhood income quintile ($n$ = 3,850

[0.41%] for Ontario; $n$ = 24,067 [0.57%] for England) were categorized into the lowest income quintile, and missing area of residence information ($n$ = 151 [0.02%] for Ontario; $n$ = 22,970 [0.54%] for England) was categorized into the urban area of residence. In Ontario, these missing data are suppressed for neighbourhoods with high rates of residential instability, which are predominantly low income and urban.

Baseline maternal morbidities were measured for a 5-year lookback period using all diagnostic codes in all hospital admissions within this period to assign the Charlson comorbidity index (Deyo version) (0, 1, or 2+ comorbid conditions) and to identify hospitalizations related to (i) any psychiatric condition, (ii) addiction-related conditions, and (iii) other mental health conditions. We chose the Deyo version of the Charlson comorbidity index as it has been most widely used in maternal mortality studies and, unlike other indices, has been validated on longer-term mortality, although not in pregnant women [36,37]. S1 and S2 Tables list diagnostic codes and provide definitions for neighbourhood income quintile and urban or rural residence.

## Statistical analysis

**Descriptive statistics.** We compared mothers with an infant affected by NAS and controls within each jurisdiction. We compared baseline characteristics using Pearson's chi-squared test (categorical variables) and ANOVA (continuous variables).

**All-cause mortality.** Our primary outcome was all-cause mortality of mothers after the birth of an infant with NAS, relative to control mothers. Survival analysis for time to all-cause mortality was modelled using multivariable Cox regression, with proportionality of hazards assessed by Schoenfeld residuals and log–log plots, and included all deaths to the end of the study period. Crude and adjusted hazard ratios were produced that were adjusted only for maternal age group at delivery to describe the extent of the mortality gap between NAS mothers and controls. We explicitly did not attempt to adjust models for other covariates as we do not attempt to infer causality. This is primarily because our data reflect maternal opioid use at a single time point (delivery). Thus we are unable to examine the direction of effect for key factors such as mental illness or socioeconomic status that may either confound the association between opioid use and death or lie on the causal pathway. We initially adjusted the model for the Charlson comorbidity index but as a result of peer review do not report these estimates because of our aim to not make causal inferences. Crude and adjusted survival curves were plotted to estimate the absolute risk of mortality at 5 and 10 years after birth. We derived 95% confidence intervals for mortality at 5 and 10 years after delivery through log–log transformation of the survival function and computed $p$-values using the $z$ test [38]. Mothers with missing values for maternal age at delivery were excluded from the adjusted models. Individuals surviving beyond the end of the study period were censored at March 31, 2016.

Age-standardized all-cause mortality rates were estimated for England and Ontario using the direct method of standardization and the Canadian 2006 Census as the standard population. We present age-standardized all-cause mortality rates stratified by maternal age at delivery (excluding those with missing age information), neighbourhood income quintile (Q1 [most deprived] versus Q2–Q5 due to low numbers of deaths in the most affluent quintiles), urban/rural residence, previous addiction-related or other mental-health-related hospitalization, Charlson comorbidity index (0 versus 1+ due to low numbers of deaths in the 2+ category for NAS mothers), and infant discharge to social service out-of-home care.

**Cause-specific mortality.** Ten-year cumulative incidence of death was calculated for each cause of death category with consideration of other causes of death as competing risk events using Gray's test for the homogeneity of 2 or more cumulative incidence functions. Individuals

surviving beyond the end of the study period were censored at December 31, 2014, for this particular analysis.

Statistical analyses were performed using Stata version 15 for England data and SAS version 9.4 statistical software for Ontario data in a Unix environment. $p$-Values for age-standardized all-cause mortality and cause-specific mortality were derived using the $z$ test.

ICES is a prescribed entity under section 45 of Ontario's Personal Health Information Protection Act. Section 45 authorizes ICES to collect personal health information, without consent, for the purpose of analysis or compiling statistical information with respect to the management of, evaluation or monitoring of, the allocation of resources to, or planning for all or part of the health system. Projects conducted under section 45, by definition, do not require review by a research ethics board. This project was conducted under section 45, and approved by the ICES Privacy and Legal Office. Research ethics approval was granted by the Hospital for Sick Children Research Ethics Board for Ontario analyses. The English analyses were exempt from UK NHS Research Ethics Committee approval because it involved the analysis of de-identified administrative data.

## Results

### Cohort characteristics of mothers in England and Ontario

After applying our inclusion criteria, there were 13,577 and 4,966 mothers of infants with NAS and 4,205,675 and 929,985 control mothers in England and Ontario, respectively. Baseline characteristics are described in Table 2. In both jurisdictions, the majority of mothers had no previous recorded hospital birth (lookback to April 1, 1997) and lived in urban areas. Compared to controls, a larger proportion of mothers of infants with NAS lived in neighbourhoods in the lowest income quintile (37.3% versus 23.5% in England and 44.9% versus 22.9% in Ontario; $p < 0.001$ for both jurisdictions); mothers of infants with NAS were on average younger than controls, and, in Ontario, a higher proportion were teenage mothers (9.8% versus 3.7%; $p < 0.001$). Mothers of infants with NAS were more likely to have higher comorbidity scores (1 or 2+ on the Charlson comorbidity index) than controls (21.2% versus 6.9% in England and 7.6% versus 1.7% in Ontario; $p < 0.001$ for both jurisdictions), and a greater proportion also had a previous psychiatric hospitalization (13.5% versus 0.6% in England and 15.3% versus 1.1% in Ontario; $p < 0.001$ for both jurisdictions). Infant discharge to care by social services was much more common among infants with NAS than among control infants in both England (9.7% versus 0.1%; $p < 0.001$) and Ontario (15.2% versus 0.1%; $p < 0.001$). We found no differences in neonatal mortality between infants with NAS and controls in Ontario, but a marginally higher rate among infants with NAS in England (0.3% versus 0.2% $p = 0.01$).

### Risk of death among mothers of infants born with NAS

From 2002 to 2016, there were 112,890 total person-years of follow-up for mothers of infants with NAS in England (34.9 million for controls) and 35,740 total person-years of follow-up for mothers of infants with NAS in Ontario (7.9 million for controls). The mean duration of follow-up was 8.4 years (England) and 7.2 years (Ontario) for NAS mothers and 8.2 years (England) and 8.6 years (Ontario) for controls. Crude mortality rate for mothers of infants with NAS was 5.01 (95% CI 4.62–5.44) per 1,000 person-years in England and 4.28 (95% CI 3.63–5.02) per 1,000 person-years in Ontario. In both jurisdictions, the crude cumulative mortality incidence (superimposed on the survival curves in Fig 1) was significantly different between mothers of infants with NAS and controls. In England, 5- and 10-year mortality (95% CI) was 1.81% (1.59%–2.05%) and 5.13% (4.69–5.62%), respectively, for mothers of infants

**Table 2. Characteristics of mothers and infants at baseline, April 1, 2002 to December 31, 2012.**

| Characteristic | England | | Ontario | |
|---|---|---|---|---|
| | NAS cases (n = 13,577) | Controls (n = 4,205,675) | NAS cases (n = 4,966) | Controls (n = 929,985) |
| **Mean (SD) maternal age at delivery, years***  | 28.5 (5.6) | 29.8 (6.1) | 27.0 (6.1) | 30.2 (5.6) |
| **Maternal age at delivery, categorized***  | | | | |
| ≤19 years | 775 (5.7) | 258,200 (6.1) | 488 (9.8) | 34,414 (3.7) |
| 20–34 years | 10,712 (78.9) | 3,041,112 (72.3) | 3,815 (76.8) | 683,910 (73.5) |
| 35+ years | 1,869 (13.8) | 898,110 (21.4) | 663 (13.4) | 211,661 (22.8) |
| Missing | 221 (1.6) | 8,253 (0.2) | 0 (0.0) | 0 (0.0) |
| **Time since last birth***  | | | | |
| No previous birth since 1997 | 6,944 (51.1) | 2,692,743 (64.0) | 2,322 (46.8) | 568,109 (61.1) |
| <2 years | 1,504 (11.1) | 315,454 (7.5) | 697 (14.0) | 78,540 (8.4) |
| 2 to 5 years | 2,761 (20.3) | 840,237 (20.0) | 1,366 (27.5) | 238,227 (25.6) |
| 6+ years | 2,368 (17.4) | 357,240 (8.5) | 581 (11.7) | 45,109 (4.9) |
| **Neighbourhood income quintile*†**  | | | | |
| Q1 (lowest) | 5,059 (37.3) | 988,493 (23.5) | 2,231 (44.9) | 212,869 (22.9) |
| Q2 | 3,487 (25.7) | 898,448 (21.4) | 1,043 (21.0) | 187,607 (20.2) |
| Q3 | 2,304 (17.0) | 792,989 (18.9) | 668 (13.5) | 188,884 (20.3) |
| Q4 | 1,622 (11.9) | 747,224 (17.8) | 599 (12.1) | 189,978 (20.4) |
| Q5 (highest) | 1,105 (8.1) | 778,521 (18.5) | 425 (8.6) | 150,647 (16.2) |
| Q2–Q5 | 8,518 (62.7) | 3,217,182 (76.5) | 2,735 (55.1) | 717,116 (77.1) |
| **Area of residence*‡**  | | | | |
| Urban | 12,217 (90.0) | 3,573,814 (85.0) | 4,164 (83.9) | 841,133 (90.4) |
| Rural | 1,360 (10.0) | 631,861 (15.0) | 802 (16.1) | 88,852 (9.6) |
| **Charlson comorbidity index***  | | | | |
| 0 | 10,695 (78.8) | 3,915,844 (93.1) | 4,586 (92.3) | 914,048 (98.3) |
| 1 | 2,606 (19.2) | 272,330 (6.5) | 304 (6.1) | 12,325 (1.3) |
| 2+ | 276 (2.0) | 17,501 (0.4) | 76 (1.5) | 3,612 (0.4) |
| **Any psychiatric condition***  | 1,832 (13.5) | 26,108 (0.6) | 760 (15.3) | 10,440 (1.1) |
| Addiction-related*  | 1,396 (10.3) | 8,922 (0.2) | 318 (6.4) | 1,411 (0.2) |
| Other mental health*  | 566 (4.2) | 18,055 (0.4) | 572 (11.5) | 9,490 (1.0) |
| **Cesarean delivery** | 3,121 (23.0)*  | 1,044,111 (24.8) | 1,374 (27.7)[NS] | 268,553 (28.9) |
| **Pre-eclampsia/eclampsia** | 290 (2.1)*  | 127,753 (3.0) | 60 (1.2)[NS] | 11,390 (1.2) |
| **Gestational diabetes***  | 48 (0.4) | 36,276 (0.9) | 181 (3.6) | 48,893 (5.3) |
| **Gestational hypertension** | 220 (1.6)*  | 123,908 (3.0) | 193 (3.9)[NS] | 40,505 (4.4) |
| **Gestational age at delivery***  | | | | |
| <34 weeks | 671 (4.9) | 92,155 (2.2) | 318 (6.4) | 18,825 (2.0) |
| 34 to 36 weeks | 1,224 (9.0) | 132,599 (3.2) | 721 (14.5) | 50,529 (5.4) |
| 37+ weeks | 7,538 (55.5) | 2,962,970 (70.5) | 3,875 (78.0) | 859,353 (92.4) |
| Missing§  | 4,144 (30.5) | 1,018,010 (24.2) | 52 (1.1) | 1,278 (0.1) |
| **Infant discharged to social services***  | 1,316 (9.7) | 4,446 (0.1) | 753 (15.2) | 1,265 (0.1) |
| **Neonatal mortality** | 37 (0.3) | 7,497 (0.2) | 18 (0.4)[NS] | 2,724 (0.3) |

Data are number (percent) unless otherwise indicated. Neonatal mortality was statistically significantly different between cases and controls for England, at $p = 0.01$. $p$-Values for continuous variables were derived from ANOVA, while $p$-values for categorical variables were derived from Pearson's chi-squared test.

*There was a statistically significant difference between cases and controls ($p < 0.001$).

†Missing values for neighbourhood income quintile for England (0.82% and 0.57% of cases and controls, respectively) and Ontario (2.11% and 0.40% of cases and controls, respectively) were included in the lowest quintile (Q1).

‡Missing values for area of residence for England (1.1% of cases and 0.54% of controls) and Ontario (0.08% and 0.02% of cases and controls, respectively) were included in the urban residence category.

[NS]There was no statistically significant difference between cases and controls.

§Gestational age is not a mandatory field reported to NHS Digital in England.

NAS, neonatal abstinence syndrome.

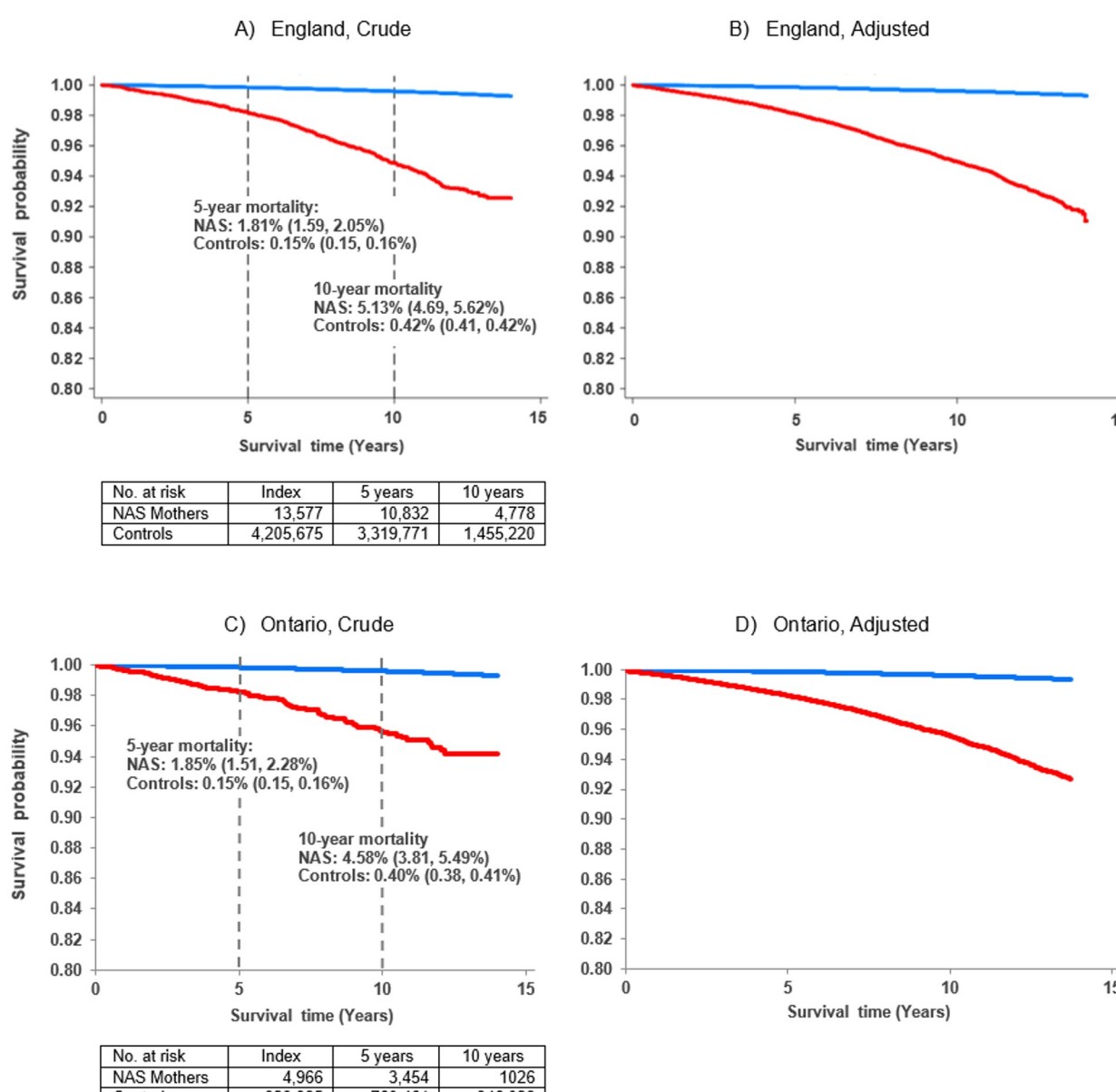

**Fig 1. Survival curves for all-cause mortality—Crude (Kaplan–Meier curve) and adjusted (derived from Cox model) for maternal age at delivery, 2002 to 2016.** Crude curves for England (A) and Ontario (C); adjusted curves for England (B) and Ontario (D). Red, neonatal abstinence syndrome (NAS) mothers; blue, controls. The adjusted survival curve for the England controls is a 10% sample of the full control population. $p < 0.001$ for a difference in all-cause mortality between NAS mothers and controls for the overall study period (crude and adjusted), 5-year mortality (crude), and 10-year mortality (crude) for England and Ontario. Mortality rates superimposed on figures are accompanied by 95% confidence limits presented in parentheses.

with NAS and 0.15% (0.15%–0.16%) and 0.42% (0.41–0.43%), respectively, for controls ($p < 0.001$ for 5- and 10-year mortality for NAS mothers versus controls); in Ontario, it was 1.85% (1.51%–2.28%) and 4.58% (3.81%–5.49%), respectively, for mothers of infants with NAS and 0.15% (0.15%–0.16%) and 0.40% (0.38%–0.41%), respectively, for controls ($p < 0.001$ for 5- and 10-year mortality for NAS mothers versus controls). The decline in survival of mothers of infants with NAS over time was steady in both jurisdictions, with no clear inflection point or distinct period of risk.

**Table 3. Age-standardized all-cause mortality rates per 1,000 women, April 1, 2002, to March 31, 2016.**

| Characteristic | England | | | | | *p*-Value | Ontario | | | | | *p*-Value |
|---|---|---|---|---|---|---|---|---|---|---|---|---|
| | NAS mothers | | Controls | | | | NAS mothers | | Controls | | | |
| | Number | Rate (95% CI) | Number | Rate (95% CI) | | | Number | Rate (95% CI) | Number | Rate (95% CI) | | |
| **Overall** | 566 | 41.7 (37.9–45.4) | 14,356 | 3.5 (3.4–3.6) | | <0.001 | 153 | 33.3 (27.6–39.9) | 3,194 | 3.6 (3.5–3.8) | | <0.001 |
| **Maternal age at delivery, years** | | | | | | | | | | | | |
| ≤19 | 11 | 15.5 (5.1–25.8) | 682 | 2.7 (2.5–2.9) | | <0.001 | 7 | 11.9 (3.5–29.1) | 167 | 5.1 (4.3–6.1) | | 0.3 |
| 20–34 | 439 | 40.5 (36.7–44.2) | 8,796 | 2.8 (2.8–2.9) | | <0.001 | 119 | 31.9 (26.3–38.3) | 2,067 | 3.0 (2.9–3.2) | | <0.001 |
| 35+ | 116 | 65.4 (50.5–80.2) | 4,878 | 6.3 (5.8–6.2) | | <0.001 | 27 | 39.5 (22.7–64.1) | 960 | 5.2 (4.8–5.6) | | <0.001 |
| **Neighbourhood income quintile** | | | | | | | | | | | | |
| Q1 (lowest) | 228 | 47.6 (40.8–54.3) | 3,801 | 4.5 (4.3–4.6) | | <0.001 | 71 | 37.7 (28.2–49.4) | 932 | 4.6 (4.3–5.0) | | <0.001 |
| Q2–Q5 | 328 | 38.3 (33.8–42.9) | 10,555 | 3.3 (3.2–3.4) | | <0.001 | 82 | 30.7 (23.7–39.1) | 2,262 | 3.4 (3.2–3.5) | | <0.001 |
| **Area of residence** | | | | | | | | | | | | |
| Urban | 526 | 43.1 (39.0–47.2) | 12,176 | 3.5 (3.4–3.6) | | <0.001 | 130 | 33.6 (27.5–40.8) | 2,728 | 3.4 (3.3–3.6) | | <0.001 |
| Rural | 40 | 28.5 (18.7–38.3) | 2,180 | 3.4 (3.2–3.5) | | <0.001 | 23 | 30.0 (16.7–49.8) | 466 | 5.4 (4.9–6.0) | | <0.001 |
| **History of psychiatric hospitalizations** | | | | | | | | | | | | |
| Addiction-related | 109 | 78.6 (62.5–94.7) | 327 | 56.8 (50.2–63.4) | | 0.01 | 16 | 58.0 (31.4–98.1) | 46 | 39.0 (27.5–53.6) | | 0.3 |
| Other mental health | 41 | 64.7 (43.9–85.6) | 358 | 20.1 (17.9–22.2) | | <0.001 | 29 | 55.3 (35.5–82.2) | 139 | 16.5 (13.5–20.1) | | <0.001 |
| **Infant discharge to social services** | 77 | 58.3 (44.2–72.3) | 83 | 22.5 (17.3–27.6) | | <0.001 | 22 | 27.8 (16.0–45.0) | 30 | 32.9 (20.9–49.1) | | 0.6 |
| **Charlson comorbidity index** | | | | | | | | | | | | |
| 0 | 400 | 37.1 (33.0–41.1) | 11,933 | 3.1 (3.1–3.2) | | <0.001 | 123 | 27.8 (22.4–34.1) | 2,881 | 3.4 (3.2–3.5) | | <0.001 |
| 1+ | 166 | 56.3 (46.8–65.8) | 2,423 | 8.7 (8.3–9.1) | | <0.001 | 30 | 80.8 (52.6–118.8) | 313 | 19.3 (16.9–22.0) | | <0.001 |

Estimates were age-standardized to the 2006 Canadian population using the age groups 12–18, 19, 20–29, 30–34, 35–37, and 38–49 years for maternal age at delivery, and 12–21, 22–29, and 30+ years for the remaining characteristics.

NAS, neonatal abstinence syndrome.

Table 3 outlines age-standardized mortality rates overall and stratified by clinical and demographic characteristics for the whole study period. In both jurisdictions, rates of death were higher among mothers of infants with NAS compared to controls across most risk groups. Differences in rates were not significant (95% confidence intervals overlapped) for NAS mothers and controls for those with a history of addiction-related admission (England and Ontario) or who had an infant discharged to social services (Ontario only) or who were 19 years old or under at delivery (Ontario only).

The crude hazard ratio for all-cause mortality among mothers of infants with NAS was 12.1 (95% CI 11.1–13.2; *p* < 0.001) in England and 11.4 (95% CI 9.7–13.4; *p* < 0.001) in Ontario (Table 4). After adjustment for age, the hazard ratio increased for England and was unchanged

**Table 4. All-cause mortality risks for mothers with infants with neonatal abstinence syndrome, crude and adjusted for maternal age at delivery, 2002 to 2016.**

| Model | England | | Ontario | |
|---|---|---|---|---|
| | Hazard ratio (95% CI) | *p*-Value | Hazard ratio (95% CI) | *p*-Value |
| Crude | 12.1 (11.1–13.2) | <0.001 | 11.4 (9.7–13.4) | <0.001 |
| Adjusted for maternal age at delivery | 13.0 (11.9–14.1) | <0.001 | 11.4 (9.7–13.4) | <0.001 |

**Table 5. Cause-specific mortality per 1,000 population among mothers (10-year cumulative incidence risk), April 1, 2002, to December 31, 2014.**

| Type of deaths | England | | | | | Ontario | | | | |
|---|---|---|---|---|---|---|---|---|---|---|
| | NAS mothers | | Controls | | *p*-Value | NAS mothers | | Controls | | *p*-Value |
| | Number | Cumulative incidence (95% CI) | Number | Cumulative incidence (95% CI) | | Number | Cumulative incidence (95% CI) | Number | Cumulative incidence (95% CI) | |
| **Avoidable (excluding cancer)** | 393 | 42.9 (38.4–47.9) | 5,758 | 2.1 (2.0–2.1) | <0.001 | 93 | 30.8 (24.1–38.8) | 956 | 1.4 (1.3–1.5) | <0.001 |
| Unintentional injuries | 177 | 19.6 (16.6–23.1) | 1,132 | 0.4 (0.4–0.4) | <0.001 | 46 | 13.9 (9.7–19.3) | 285 | 0.4 (0.4–0.5) | <0.001 |
| Intentional injuries | 63 | 6.5 (4.9–8.7) | 1,298 | 0.5 (0.4–0.5) | <0.001 | 20 | 6.4 (3.8–10.4) | 312 | 0.5 (0.4–0.5) | <0.001 |
| Drug use disorders | 90 | 9.7 (7.6–12.3) | 701 | 0.3 (0.3–0.3) | <0.001 | 15 | 5.7 (3.1–9.9) | 45 | 0.1 (0.1–0.1) | <0.001 |
| All other avoidable deaths | 63 | 7.8 (5.9–10.2) | 2,627 | 0.9 (0.9–1.0) | <0.001 | 12 | 4.8 (2.3–9.1) | 314 | 0.5 (0.4–0.5) | <0.001 |
| **Unavoidable (excluding cancer)** | 42 | 4.2 (3.0–5.9) | 1,649 | 0.6 (0.5–0.6) | <0.001 | 19 | 8.5 (4.6–14.7) | 417 | 0.6 (0.6–0.7) | <0.001 |
| **Cancer (avoidable and unavoidable)** | 16 | 1.6 (1.5–1.7) | 4,233 | 1.6 (0.9–2.8) | 1.00 | 10 | 3.3 (1.6–6.3) | 1,113 | 1.8 (1.7–1.9) | 0.2 |
| **Missing cause[a]** | ≤10 | — | ≤10 | — | — | ≤ 5 | — | 139 | 0.2 (0.2–0.2) | — |

[a]Privacy legislation requires suppression of cell sizes <11 in England and <6 in Ontario.

NAS, neonatal abstinence syndrome.

for Ontario (England adjusted HR 13.0; 95% CI 11.9–14.1; *p* < 0.001; and Ontario adjusted HR 11.4; 95% CI 9.7–13.4; *p* < 0.001).

Table 5 presents results for cause-specific mortality between 2002 and 2014 (cause-specific data for 2015–2016 were not available in Ontario). Avoidable deaths were the most common cause of death among mothers of infants with NAS in both jurisdictions (accounting for >85% in England and 75% in Ontario), with a 10-year cumulative incidence risk of 42.9 deaths per 1,000 population (95% CI 38.4–47.9) among English mothers and 30.8 deaths per 1,000 population (95% CI 24.1–38.8) among Ontario mothers. Intentional and unintentional injuries (e.g., transport injuries, unintentional falls) made up the majority of avoidable mortality in the mothers of infants with NAS in both jurisdictions.

## Discussion

In this large population-based study across 2 countries, 1 in 20 mothers of infants with NAS died within 10 years of delivery—a mortality risk that was 11–12 times higher than for control mothers. Findings were consistent across both jurisdictions. For virtually all causes of death, mortality rates were substantially higher for mothers of infants with NAS than for controls,

with the majority of deaths attributable to avoidable causes such as intentional and unintentional injuries. We also identified universally high mortality rates among mothers who had a history of hospitalization for addiction, irrespective of whether or not their infant had NAS. We found no evidence of a high risk period in the 1–2 years after birth (corresponding to the period typically targeted by public health nursing or other support for high-risk families) for maternal deaths in the NAS group.

Other population-based studies report increased perinatal maternal mortality in mothers using opioids [5,16,17] and longer-term risk in mothers with alcohol or drug misuse during pregnancy [20,22] (Table 1). None of these studies address long-term all-cause mortality for mothers of NAS-affected infants as we have done. High mortality rates have been reported for non-pregnant populations of female opioid users. Notably, 3 large-scale studies present crude mortality rates per 1,000 person-years of 6.5 (95% CI 6.1–6.9) in New South Wales [39], between 7.5 and 13.9 in opioid-using women aged 15–44 years in California [29], and 12.2 (95% CI 10.3–14.4) and 19.7 (95% CI 15–25.8), respectively, for female users of heroin and other opioids in Denmark [40]. Our crude mortality rates per 1,000 person-years for mothers with infants with NAS of 5.01 (95% CI 4.62–5.44) in England and 4.28 (95% CI 3.63–5.02) in Ontario are comparatively lower, which is likely driven in part by a relatively younger age distribution in our study population. For example, in the New South Wales study, Degenhardt et al. included all ages and reported increasing mortality rates with older age. The higher risk of all-cause mortality we report is not surprising and has been shown in other marginalized groups or those with mental health problems [21,41]. In particular, unintentional injury deaths (which include those related to victimization) predominate and may result from social vulnerability or misclassification of injuries that are intentional [42–46].

In our study, the high rate of premature mortality among the mothers of infants with NAS was mirrored by high rates for the mothers of control infants discharged to social services and for mothers with a history of hospitalization for addiction. Other studies have demonstrated maternal well-being declining in association with the loss of a child to foster care [42,47], whereas retaining care of the child may help facilitate treatment [48]. An estimated 7%–20% of NAS-affected infants do not return home with their mother at the time of postnatal discharge from hospital [49–52], which is similar to the percentage in our study (10%–15%). However, these figures may not reflect the much higher cumulative risk of foster care placements occurring later in childhood [53]. Our findings also mirror the growing body of literature describing the constellation of psychosocial risk factors linking mental illness, addiction, and social adversity [43,54] and suggest the need for multifaceted support for these mothers irrespective of whether their children are living with them.

The longitudinal and population-based nature of this study, its size, and comparison of similar universal healthcare systems are strengths. Limitations include potential linkage error, misclassification of mothers using opioids whose babies did not develop NAS, and lack of direct measures of maternal opioid use or treatment, or other substance misuse, which may underestimate the burden of mortality in mothers with opioid use within our study or make findings less generalizable to opioid-using mothers whose infants do not have NAS or who live in other jurisdictions with other approaches to treatment and available supportive services. In the extract of English hospital data, 96% of live births were matched to maternal records, but linkage was lower (88%) for infants with NAS. Comparison of the characteristics of these English infants with NAS by linkage status indicated an association between non-linkage and both longer hospital stays and greater risk of placement in out-of-home care [55], with the implication that our results may underestimate the risk of death among women with opioid use. Our results reflect the subset of mothers with prenatal opioid exposure who gave birth to an infant with NAS, and as such they likely underestimate the true risk of maternal mortality

associated with prenatal opioid use. However, we also describe mortality in mothers without an infant with NAS, but who received care for mental illness and addictions, thus broadening the scope for generalizability. Our study cohort may also include the rare cases of NAS related to withdrawal from other substances [56] or from postnatal opioid use ("iatrogenic NAS") [57], but data on these other exposures were not available.

Our study has implications for research, practice, and policy to improve maternal and, arguably, child outcomes related to prenatal opioid use. Enhanced treatment programs for opioid dependence that integrate maintenance therapy, psychotherapy, reproductive health, and obstetric care have been found to be effective in reducing substance misuse, unplanned pregnancies, and obstetric complications during the perinatal period [58]. Some evaluations of programs supporting mothers with opioid use and their children suggest that multifaceted services addressing health, addiction, housing, and parenting needs can improve parenting capacity and attachment and reduce child apprehension [56,59–64]. However, rigorous evidence on interventions promoting long-term support is limited and should be a research and policy priority. New funding for child welfare agencies in the US to provide services related to mental health, addictions, and parenting in response to the growing numbers of mothers using opioids is an opportunity for evaluation of different models of support [65]. Most current home-visiting programs target only families with children and only for a short period of time. Our findings suggest that interventions need to extend past the early postpartum period and include mothers whose children may not return home. Finally, the findings from our study also indicate that studies and surveillance focused only on deaths directly attributable to opioid overdose will miss the full extent of the problem, given the importance of deaths due to unintentional and intentional injuries, not all of which involve opioids.

In conclusion, while much attention and research on NAS has focused on infant and child outcomes in isolation, our study is the first population-based analysis to our knowledge of long-term maternal mortality following the birth of an infant with NAS. The findings provide a stark reminder of the vulnerability and sustained poor outcomes of these mothers. Policy responses to the current opioid epidemic will require effective strategies for risk mitigation and ongoing support for families affected by opioid use. Large-scale linkage of health and social care administrative data would facilitate ongoing research, program evaluation, and surveillance.

## Supporting information

**S1 Table. Study diagnostic codes and description of sociodemographic characteristics.**
(DOCX)

**S2 Table. Description of baseline sociodemographic characteristics for neighbourhood income quintile and urban and rural area of residence.**
(DOCX)

**S1 Text. Dataset creation plan and analytic plan: Long-term mortality in mothers of infants with neonatal abstinence syndrome: A population-based parallel-cohort study in England and Ontario, Canada.**
(DOCX)

**S2 Text. STROBE Statement—Checklist of items that should be included in reports of cohort studies.**
(DOCX)

## Acknowledgments

The opinions, results, and conclusions reported in this paper are those of the authors and are independent from the funding sources. **Ontario**: No endorsement by ICES or the Ontario Ministry of Health and Long-Term Care is intended or should be inferred. Parts of this material are based on data and information compiled and provided by the Canadian Institute for Health Information (CIHI) and the Ontario Office of the Registrar General (ORG), the original source of which is ServiceOntario. However, the analyses, conclusions, opinions, and statements expressed herein are those of the authors, and not necessarily those of CIHI, ORG, or the Ministry of Government Services. **England**: This article represents independent research, and the views expressed are those of the authors and do not represent those of the NHS, the National Institute for Health Research, or the Department of Health and Social Care.

## Author Contributions

**Conceptualization:** Astrid Guttmann, Ruth Blackburn, Ruth Gilbert.

**Data curation:** Ruth Blackburn, Limei Zhou.

**Formal analysis:** Ruth Blackburn, Limei Zhou.

**Funding acquisition:** Astrid Guttmann, Ruth Gilbert.

**Investigation:** Astrid Guttmann, Ruth Blackburn, Abby Amartey, Limei Zhou, Linda Wijlaars, Natasha Saunders, Katie Harron, Maria Chiu, Ruth Gilbert.

**Methodology:** Astrid Guttmann, Ruth Blackburn, Limei Zhou, Ruth Gilbert.

**Project administration:** Astrid Guttmann, Ruth Blackburn, Abby Amartey, Ruth Gilbert.

**Supervision:** Astrid Guttmann, Ruth Gilbert.

**Visualization:** Ruth Blackburn, Limei Zhou.

**Writing – original draft:** Astrid Guttmann, Ruth Blackburn, Abby Amartey, Limei Zhou, Ruth Gilbert.

**Writing – review & editing:** Astrid Guttmann, Ruth Blackburn, Abby Amartey, Limei Zhou, Linda Wijlaars, Natasha Saunders, Katie Harron, Maria Chiu, Ruth Gilbert.

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
