## [Decision Letter · Decision Letter 0]

22 Aug 2019

Dear Dr. Guttmann,

Thank you very much for submitting your manuscript "Long-term mortality in mothers of infants with neonatal abstinence syndrome: Findings from a population-based parallel-cohort study in England and Ontario, Canada" (PMEDICINE-D-19-02772) for consideration at PLOS Medicine. 

[LINK]

In light of these reviews, I am afraid that we will not be able to accept the manuscript for publication in the journal in its current form, but we would like to consider a revised version that addresses the reviewers' and editors' comments. Obviously we cannot make any decision about publication until we have seen the revised manuscript and your response, and we plan to seek re-review by one or more of the reviewers. 

We consider that your paper may be suitable for inclusion in our upcoming Special Issue on Substance Use, Misuse and Dependence, and we may be able to include your paper in this, all being well. Please indicate in your rebuttal letter whether this would be of interest to you. 

We expect to receive your revised manuscript by Sep 05 2019 11:59PM. Please email us (plosmedicine@plos.org) if you have any questions or concerns.

We look forward to receiving your revised manuscript. 

Sincerely,

Louise Gaynor, MBBS PhD

Associate Editor 

PLOS Medicine

plosmedicine.org

Title

Please revise your title according to PLOS Medicine's style. ‘Findings from’ can be omitted.

Abstract 

Abstract Background: 

Please remove the hyphens between opioid and addiction / use. 

Abstract Methods and Findings:

Please add summary demographic details for study participants early in the "methods and findings" subsection of your abstract.

Please clarify why death records only extended to 10 years after delivery when health data was assessed from 2002 and death records assessed up until 2016

Please ensure that all numbers presented in the abstract are present and identical to numbers presented in the main manuscript text.

Please clarify what is meant by ‘index delivery’ 

Please include 95% CIs and p values for 10-year mortality data.

Please clarify which of the data presented are observed or modelled

Please include the important dependent variables that are adjusted for in the analyses

Please expand upon what was considered an ‘avoidable cause’ of death

Abstract Conclusions:

Please add the main finding from the control group for comparison. You state in the first line of your discussion that ‘a mortality risk that is 11-12 times higher than for control mothers’, for example.

Please interpret the study based on the results presented in the abstract, emphasizing what is new without overstating your conclusions. 

Please begin the first sentence of the "conclusions" subsection of your abstract with "In this study, we found that ..." or similar.

Author Summary

Introduction 

Please explain the need for and potential importance of your study. 

If there has been a systematic review of the evidence related to your study (or you have conducted one), please refer to and reference that review and indicate whether it supports the need for your study. 

Please conclude the Introduction with a clear description of the study question or hypothesis, including what is measured as primary and secondary outcomes.

Methods

Did your study have a prospective protocol or analysis plan? Please state this (either way) early in the Methods section. If available, please include the relevant prospectively written document with your revised manuscript as a Supporting Information file to be published alongside your study, and cite it in the Methods section. A legend for this file should be included at the end of your manuscript. If no such document exists, please make sure that the Methods section transparently describes when analyses were planned, and when/why any data-driven changes to analyses took place. In either case, changes in the analysis-- including those made in response to peer review comments-- should be identified as such in the Methods section of the paper, with rationale. 

Please indicate: (1) the specific hypotheses you intended to test, (2) the analytical methods by which you planned to test them, (3) the analyses you actually performed, and (4) when reported analyses differ from those that were planned, transparent explanations for differences that affect the reliability of the study's results. 

Please clarify what is meant by ‘Mothers and infants and these different datasets were linked’

Please clarify what is meant by ‘randomly selected one delivery date’

Please clarify what is meant by ‘index delivery’ 

Please ensure that the study is reported according to the STROBE guideline. Please add the following statement, or similar, to the Methods: "This study is reported as per the Strengthening the Reporting of Observational Studies in Epidemiology (STROBE) guideline (S1 Checklist).”

Results

Where results are presented in the cohort characteristics, please quantify the differences between mothers of infants with NAS and controls with % and p values. (page 9)

On page 9, please clarify what you mean by "significant". If statistical significance is intended, please indicate that. 

When a p value is given, please specify the statistical test used to determine it.

Please avoid general terms like ‘similar’ and ‘generally’

Please include 95% CIs and p values for 5-year and 10-year mortality data.

Please indicate which factors are adjusted for in the main text (page 12) 

The Supporting Information file Table S3 is central to the understanding of the paper. Please incorporate it into the main paper (page 13).

Please clarify why data not available up to 2016 (page 14).

Please highlight in your results section the data for which you infer that ‘We also identified universally high mortality rates among mothers who had a history of hospitalization for mental health or addictions, irrespective of whether or not their infant had NAS.’ in your discussion.

Discussion

Please present and organize the Discussion as follows: a short, clear summary of the article's findings; what the study adds to existing research and where and why the results may differ from previous research; strengths and limitations of the study; implications and next steps for research, clinical practice, and/or public policy; one-paragraph conclusion.

Please clarify what is considered a ‘ early high risk period’ (p.16)

PLOS does not permit "data not shown” (page 17). Please remove this claim, or do one of the following:

a) If you are the owner of the data relevant to this claim, please provide the data in accordance with the PLOS data policy, and update your Data Availability Statement as needed.

b) If the data not shown refer to a study from another group that has not been published, please cite personal communication in your manuscript text (it should not be included in the reference section). Please provide the name of the individual, the affiliation, and date of communication. The individual must provide PLOS Medicine written permission to be named for this purpose.

c) For any other circumstance, please contact me ASAP.

References

In your reference list, please ensure that journal names are abbreviated consistently, and add spaces where needed (e.g. reference 32 "BrMedBull").

Tables

Table 2 - please indicate which differences between NAS cases and controls are statistically significant (or are not, if this is easier) for ease of interpretation. The table legend is lengthy and some text could be replaced by * and NS annotations within the table itself. 

Figures

In Figure 1, please provide 95% CIs, and please indicate whether differences between 5-year mortality are statistically significant.

Comments from the reviewers:

Reviewer #1: This article is about the long term mortality of mothers likely to have taken opioids during pregnancy as indicated by neonatal abstinence syndrome of the newborn children. In general, the article is well written, uses a large and detailed data set, and uses methods suitable for analysis throughout. Other reviewers will be better placed to comment on the novelty of the findings and how this could affect any clinical practice, but there are nonetheless some comments that I think could be addressed by the authors:

(1) On page 4 of introduction (no line numbers) it is stated that "risks may differ given patterns of opioid use in the current global epidemic" but I don't think the patterns of use is tested within this study, and I don't believe that there is evidence within report that the effect of opioid use would differ depending on the population prevalence/patterns being assessed as opposed to the effects on an individual. 

(2) The study population of mothers seems to be relevant only in terms of the children providing a proxy measure of the opioid status, and that as stated on page 10 there was "no difference in neonatal mortality between infants with NAS and controls". As such, it may be relevant to expand further on how comparable the results from this study are in terms of mortality rates to other studies that exist and have taken a more general population but with different issues of accurately recording opioid use as suggested in introduction (eg lack of prescription data, measuring illicit use). 

(3) Further detail is in my opinion necessary to justify the non-consideration of other covariates in the statement of statistical analysis section that "..other covariates may be on the causal pathway". While this could be a justification for excluding terms as adjustments in the model, there is no evidence within the report of the proposed directions of causality (eg in graph), and I would consider that some of the other available covariates (eg deprivation status) could potentially be a confounding factor as opposed to being caused in the first instance by opioid use. 

(4) Given the categorisation of age and charlson score in some of the descriptive tables, it would be useful to confirm if in fact that continuous measures are used for the models that have been fitted as this could provide a more appropriate model than if they have been categorised in a relatively arbitrary way. 

(5) Individuals would technically not be "censored at death" as for all cause-mortality death is the outcome/event of interest and there would be no time after death to censor (as there would be with the end of follow up)

(6) To asses rates by factors not in the models stratified analysis has been done, but this is without first having tested interaction terms of the terms under consideration and opioid group which would provide more robust interpretation of whether there are any genuine sub-group differences.

(7) In figure 1 results from adjusted models are shown in Kaplan Meier plots, but it is unclear exactly what results this is showing. Potentially it is for the average age and average charlson score across the entire study, and if so this could be made clear within the figure caption. 

(8) The grouping of Q2-Q5 for neighborhood income quarintile in table 3 seems somewhat arbitrary instead of using the quintiles directly, particularly as missing data has been recoded into the lowest level which could also have an effect on how this comparison would be interpreted. Although the assumptions for recoding the missing income-quintile or rural/urban in general seems reasonable, the robustness of the results to these assumptions and whether there is any systematic reasons for why there would be missing data if not just random would be useful to comment on. 

(9) In the discussion section on page 16, it is noted that there is "potential for impact to improve maternal and child outcomes", and similarly on page 18 "potential implications for their children", but within the analysis done as part of this article it is only stated that there is no difference in mortality between NAS and control children in terms of mortality and no other child outcomes were considered or summaries shown. Based on the data from this study alone the statements on child outcomes are therefore not supported. However, there are also summaries in the discussion section relating to children not presented in the main results of the study ("..10-15% NAS affected infants do not return home with mother..", "greater risk of placement in out of home care") suggesting that there potentially is data available that could be shown to support the statement and give greater clarity on the outcomes for children. 

Reviewer #2: Thank you for the opportunity to review this interesting manuscript.

1. I was a little confused about the study period and the look-back time, when reading through the first time. Was there linked hospital data back to 1 April 1997 for all? Was the look-back a consistent 5 years for all, or was it back to 1997 for all? All-cause mortality had data up to March 31 2016, so only those from 2002 to 2006 had a full 10 years of follow-up, is that correct? Was 10 years the maximum follow-up time for each individual? Please add further details in the methods. Please specify the average follow-up time per person (split by jurisdiction as well as NAS and controls) in the Results section.

2. The abstract background discusses the pregnancy and delivery settings as an opportunity to access support, and potentially assist new mothers with opioid addiction. However, I couldn't see mention of this in the Introduction. If this is an important theme of the paper, then I think it needs further exploration. What, if any, studies have shown an impact of targeting pregnant women or mothers with opioid addiction? Are women on methadone or other opioid-replacement therapy better off than those who are not? 

3. Please state the number and proportion of all the missing data noted in the methods, e.g. income, area of residence and age. The authors state that missing values for neighbourhood income quintile were put in the lowest income quintile, and that missing area of residence was categorised into urban. Including the missing data in one category will bias your findings towards the null, as the missing presumably include a proportion of values in all categories. I would suggest either removing the missing, if the proportion is very small, or using principled missing data methods such as multiple imputation or full information maximum likelihood, if the missing is more substantial. And why is the missing data for gestational week of birth so high in the English cohort?

4. The Charlson Comorbidity index is a weighted score based on the relative risk of mortality. The index is not a continuous variable, nor a count of comorbidities (please revise the footnote on Table 2). Using it as a continuous variable (e.g. reporting mean and including it in the Cox regression as a continuous variable) suggests that the relative impact on mortality from 0 to 1 comorbidity score is the same as that from 4 to 5, for example. What is the justification for using it as a continuous variable? Has the functional form been tested and shown to be linear?

5. I agree that it is important to control for other underlying causes of death, not related to the possible opioid addiction, but there are some aspects of the Charlson score that are probably on the causal pathway, like HIV/AIDS and liver disease. I think a directed acyclic graph of the theoretical relationships between the variables and the mortality outcome would make the assumptions about the causal pathways explicit.

Minor comments:

Page 7, para 3, line 7: Suggest removing "young", as the extreme values are young and old.

Page 7, para 3, line 8: "Missing values for neighbourhood income quintile" should be followed by "were" not "was".

Page 12, para 1, line 3: I understand the intent of this sentence but on first read, it sounds like the cumulative mortality was similar for those with NAS and controls. Suggest rewording.

Table 2 - The percentage of primiparous women in the control groups looks very high. A quick look for other studies using Ontario or Toronto data, shows the proportion of primips is between 39 to 47%, not close to 60%. (E.g. Park et al, 2015, JOGC; Booth et al, 2017, CMAJ). I did not search for English studies.

Table 2 - I would be interested in whether the preterm births are planned or spontaneous. Is it possible to split this out? 

Page 17, para 2, line 2: What is the expected impact of these limitations on the mortality outcomes?

Page 18, para 1, line 6: Are there differences between England and Ontario really a source of 'bias', if the data are never combined? 

Reviewer #3: This is a very interesting and important study looking at the relationship of NAS (using it as a marker of OUD in pregnancy) in the infant to all-cause maternal mortality in the years following delivery. It is well-executed. The strengths and limitations are well-delineated. The abstract is clear and reflects the findings in the paper. The discussion and implications to public health were well-stated.

I appreciated the summary of the introductory literature in table 1

Suggestions for revision

1. Table 2 would be strengthened by the addition of p-values to each variable.

2. Supplementary table 3 is so important. I would move it to the body of the paper.

3. Also Tables 3 and 4 would be strengthened by aOR.

[LINK]

---

## [Decision Letter · Decision Letter 1]

9 Oct 2019

Dear Dr. Guttmann,

Thank you very much for re-submitting your manuscript "Long-term mortality in mothers of infants with neonatal abstinence syndrome: A population-based parallel-cohort study in England and Ontario, Canada" (PMEDICINE-D-19-02772R1) for review by PLOS Medicine.

I have discussed the paper with my colleagues and the academic editor and it was also seen again by one of the reviewers. I am pleased to say that provided the remaining editorial and production issues are dealt with we are planning to accept the paper for publication in the journal.

*PLEASE NOTE: we have a very close deadline for inclusion in the substance abuse special issue, which publishes throughout November and so please do submit as soon as possible to allow every opportunity for this to make the deadline, but not before you hear from my colleagues in production who will be in touch in the next couple of days with formatting edits they require - please do not resubmit until you here from them too and combine all edits into one resubmission. 

[LINK]

We look forward to receiving the revised manuscript by Oct 16 2019 11:59PM. 

Sincerely,

Clare Stone 

for 

Louise Gaynor, MBBS PhD

Associate Editor 

PLOS Medicine

plosmedicine.org

Requests from Editors:

Abstract - Please add p values to all quantifiable data and where 95% Cis are provided (also throughout main text and tables)

Data statement – Ontario date – we need a point of contact that isn’t an author, per PLOS data policy. 

Author summary –please add it into the main text immediately after the abstract and remove the Supp file containing it. In addition, it’s quite long and so I suggest you remove:

• Only one recent study from the United States has explored death rates in women who took opioids in pregnancy, but solely in the first year and focused only on opioid-related deaths

And 

• Surveillance of the harm of the opioid crisis should measure all types of mortality, not just that related to opioid overdoses, and jurisdictions should capitalize on available sources of health, social and demographic administrative data to monitor and evaluate programs

In your author summary, you say “Mothers with infants with versus without NAS were more likely to be teenagers” yet table shows that iin fact age 20-34is the largest number of NAS cases for both cohorts and fololowed by age 35+ and then te4enagers. Have I misunderstood table 2? If not, please remove declamatory language from the author summary. 

Page 21 (our study is the first population-based analysis) – to our knowledge..

Comments from Reviewers:

Reviewer #2: Thank you to the authors for responding to my queries and comments. I only have one minor additional comment.

I think that the current description of the random selection of one birth for each mother still needs further clarification. As a suggestion, perhaps:

"We restricted the cohort to singleton births and if a woman had more than one livebirth delivery during the study period, one delivery was chosen at random as the focus of the study. Thus, a delivery date was selected at random and used as the study entry point for the mother (referred to as index delivery) and subsequent deliveries were ignored."

[LINK]

---

## [Editor Report · Decision Letter 2]

21 Oct 2019

Dear Dr. Guttmann, 

On behalf of my colleagues and the academic editor, Dr. Louisa Degenhardt, I am delighted to inform you that your manuscript entitled "Long-term mortality in mothers of infants with neonatal abstinence syndrome: A population-based parallel-cohort study in England and Ontario, Canada" (PMEDICINE-D-19-02772R2) has been accepted for publication in PLOS Medicine. 

PRODUCTION PROCESS

PRESS

PROFILE INFORMATION

Thank you again for submitting the manuscript to PLOS Medicine. We look forward to publishing it. 

Best wishes, 

Louise Gaynor, MBBS PhD

Associate Editor 

PLOS Medicine

plosmedicine.org